# Neutral W(V) Complexes Featuring the W$_2$O$_2$(μ-O)$_2$ Core and Amino Acids or EDTA Derivatives as Ligands: Synthesis and Structural Characterization

Hristo P. Varbanov [1,2,*] , Ferdinand Belaj [2] , Toma Glasnov [3] , Simon Herbert [4] , Thomas Brumby [4] and Nadia C. Mösch-Zanetti [2,*]

[1] Department of Pharmaceutical Chemistry, Institute of Pharmacy, University of Innsbruck, Innrain 80/82, 6020 Innsbruck, Austria
[2] Institute of Chemistry-Inorganic Chemistry, University of Graz, Schubertstraße 1/III, 8010 Graz, Austria
[3] Institute of Chemistry-Medicinal Chemistry, University of Graz, Schubertstraße 1/IV, 8010 Graz, Austria
[4] Research & Development, Pharmaceuticals Laboratory, Bayer AG, 13342 Berlin, Germany
[*] Correspondence: hristo.varbanov@uibk.ac.at (H.P.V.); nadia.moesch@uni-graz.at (N.C.M.-Z.)

**Abstract:** Multinuclear complexes of heavy metals, such as tungsten, have demonstrated considerable potential as candidates for advanced radiocontrast agents. Of particular interest is the development of stable non-ionic compounds with high metal content and reasonably low osmolality in solution. Accordingly, we have synthesized a series of neutral W(V) complexes that contain the W$_2$O$_2$(μ-O)$_2$ core and amino acids or disubstituted EDTA derivatives as ligands. The compounds were prepared from the oxalatotungstate(V) complex via a convenient procedure utilizing microwave heating. Their detailed characterization was accomplished by electrospray ionization high-resolution mass spectrometry (ESI-HRMS), $^1$H and $^{13}$C NMR spectroscopy, elemental analysis, and X-ray crystallography. Further experiments to evaluate the utility of the complexes as radiocontrast media were precluded by their poor aqueous solubility.

**Keywords:** tungsten(V); dinuclear complexes; microwave synthesis; crystal structure; CT contrast agents

## 1. Introduction

Tungsten has versatile coordination chemistry forming mono- and multinuclear complexes with different coordination numbers where it exhibits oxidation states varying between −2 and +6 [1]. Tungsten complexes have been investigated for their potential applications in catalysis [2–7], medicine [8–13], and as models for tungstoenzymes [14]. Multinuclear W(V) and W(IV) chelate complexes, for instance, represent intriguing candidates for metal-based contrast media, that can advance X-ray computed tomography (CT) [8–10]. Several compounds of this type have been evaluated in ex vivo and in vivo X-ray imaging studies and demonstrated significant improvements in image contrast and quality compared to currently used iodinated contrast agents. Prominent examples include trinuclear W$_3$S$_4$ and W$_3$O$_2$ clusters with polydentate carboxylate ligands, as well as the di-μ-oxo W(V) dimer, Na$_2$[W$_2$O$_2$(μ-O)$_2$(EDTA)] (see Figure 1a) [8–10]. Consequent clinical development of these compounds is limited so far largely due to their high osmolality in solutions, resulting in dose-limiting side effects.

The preparation of W(V) EDTA complexes featuring the W$_2$O$_2$(μ-O)$_2$ unit have been first described by Novak and Podlaha in 1974 [15]. Subsequently, the synthesis, structural characterization, and solution chemistry of other bis-anionic O,O-, S,S- and O,S-bridged W(V) dimers with EDTA and its optically active homolog 1,2-diaminopropane-N,N,N′,N′-tetraacetic acid (see Figure 1a–c) have also been reported [16–19]. Development of neutral and monoanionic analogs of these W(V) compounds would be of interest as candidates for

radiocontrast agents as they would possess lower osmolality in solution and thus fewer adverse effects by intravascular application. Accordingly, we have recently presented a ligand strategy, based on monosubstituted EDTA derivatives, that enabled the development of monoanionic di-μ-sulfido W(V) dinuclear complexes with high tungsten content and stability, and reasonably low osmolality (Figure 1d–f) [20]. Non-ionic analogs could also be obtained and structurally characterized with the amino acids histidine and 2,3-diaminopropionic acid, but not with disubstituted EDTA counterparts [20]. To the best of our knowledge, no neutral di-μ-oxo W(V) dinuclear complexes featuring amino acids or EDTA derivatives have been reported in literature so far.

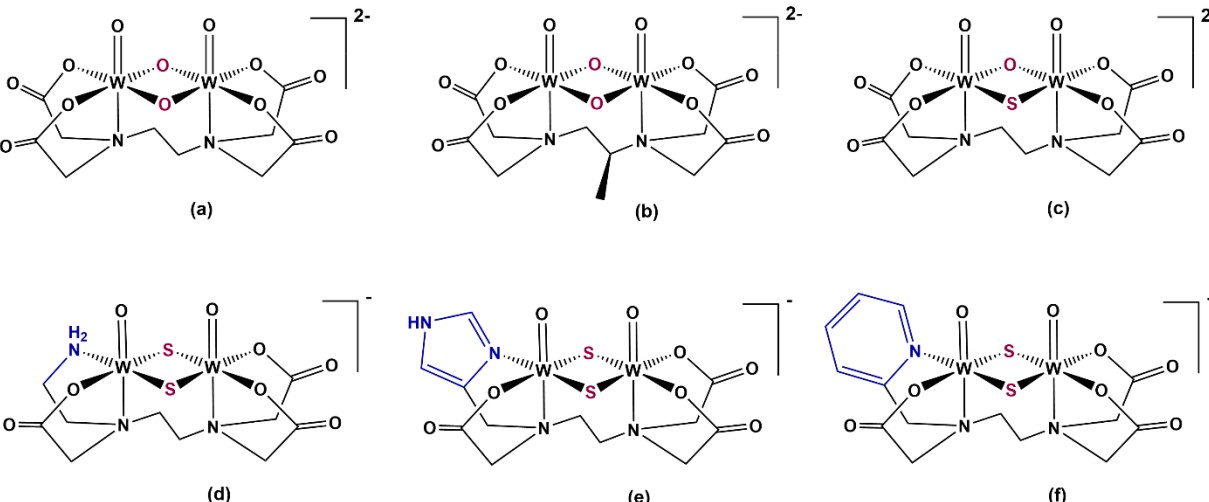

**Figure 1.** Chemical structures of bis-anionic $[W_2O_2(\mu\text{-}O)_2(EDTA)]^{2-}$ (**a**) [15], $[W_2O_2(\mu\text{-}O)_2(R\text{-}PDTA)]^{2-}$ (**b**) [17], $[W_2O_2(\mu\text{-}O)(\mu\text{-}S)(EDTA)]^{2-}$ (**c**) [18], and the monoanionic sulfido-bridged analogs (**d–f**) [20]. Counterions comprise Na$^+$ or K$^+$. EDTA: ethylenediaminetetraacetate, *R*-PDTA: (*R*)-propylenediaminetetraacetate.

Herein, we present the synthesis and structural characterization of the first examples of neutral di-μ-oxo W(V) dinuclear complexes featuring amino acids or disubstituted EDTA derivatives as ligands. The compounds were prepared from the W(V) oxalato complex (a precursor for the synthesis of $Na_2[W_2O_4(EDTA)]$) via a convenient procedure utilizing microwave heating. Their detailed characterization was accomplished by electrospray ionization high-resolution mass spectrometry (ESI-HRMS), $^1H$ and $^{13}C$ NMR spectroscopy, elemental analysis, and X-ray crystallography.

## 2. Results and Discussion

### 2.1. Synthesis

To afford neutral di-μ-oxo W(V) complexes, a set of amino acids and EDTA counterparts where two carboxylates have been exchanged to neutral coordination moieties have been explored as potential ligands (see Figure 2).

A suitable starting material for preparation of oxo-bridged W(V) dimers is the poorly characterized W(V) oxalato complex, usually formulated as $M_3[WO_2(C_2O_4)_2]$ [15,16,21] or $M_3[W_2O_4(C_2O_4)_{2.5}]$ [22] (M = Na, K or $NH_4$). The latter was prepared by Sn reduction of $Na_2WO_4$ in the presence of oxalate and oxalic acid in accordance with literature procedures [16,21,23]. Analytical data (ESI-HRMS, magnetic measurements, and elemental analysis) support a $W_2O_2(\mu\text{-}O)_2$ dimeric structure [22], rather than the suggested one by Collenberg, $Na_3[WO_2(C_2O_4)_2]$ [21] (see Figure S1 for MS spectra). We have also observed inconsistencies with the preparation and the properties of the isolated complex, similar to those described in literature [16,21,24]. For instance, the amounts of oxalate in the isolated sample variated from batch to batch; these variations have shown an effect on the kinetic stability of the compound in aqueous solutions (see Figure S2). To provide consistent

stoichiometries for subsequent reactions, the composition of the precursor was verified by elemental analyses for every individual batch.

**Figure 2.** Disubstituted EDTA derivatives and amino acids examined as potential ligands for the development of neutral di-µ-oxo W(V) dinuclear complexes. DAPA: DL-2,3-diaminopropionic acid.

The pH-dependent reactivity of the oxalatotungstate(V) complex towards the ligands depicted in Figure 2 was investigated to determine suitable reaction conditions for preparation of neutral oxo-bridged W(V) dimers. Coordination of the ligands (except for DAPA) with formation of new complexes could be accessed at pH 5–6 (evidenced by HPLC and ESI-MS measurements), while fast decomposition of the tungsten precursor was observed at pH < 3 and pH > 8. Proper reaction monitoring was difficult in some cases due to decomposition side reactions and/or low solubility of the ligand at pH 4–6. Overall, it would be beneficial to perform the synthesis under $N_2$ atmosphere and within shorter reaction times at higher temperatures in order to limit decomposition of the oxalato precursor and side products formation. In this context, microwave-assisted chemistry presents a valuable alternative to conventional heating and can be employed to facilitate preparation of new oxo-bridged W(V) dimeric complexes.

Neutral complexes of the type $[W_2O_4(L)]$, where L = L2–4 or L-histidine, could be successfully prepared starting from the W(V) oxalato precursor (**P1**) via MW-assisted ligand-exchange reaction at pH 5.5–6 and temperatures of 120–140 °C (Scheme 1). Final products were isolated after removing the oxalate formed (upon precipitation with $Ca(CH_3COO)_2$ and subsequent filtration) and crystallization at 4 °C. $[W_2O_4(L3)]$ (**3**) was obtained as an orange crystalline solid in yields over 75%. Corresponding complexes with L2, L4, and L-histidine were obtained in lower yields (<50%), in part due to decomposition side reactions. Efforts to prepare a neutral complex of the type $[W_2O_4(L1)]$ were unsuccessful, although some indications for coordination of L1 were observed in the ESI-MS spectra. RPIP-HPLC revealed formation of several species, none of which correspond to a neutral compound.

**Scheme 1.** Synthesis of neutral di-µ-oxo W(V) dinuclear complexes with L2–4 and L-Histidine. His: Histidinate.

### 2.2. Characterization

All new compounds were characterized in detail by ESI-HRMS, $^{1}$H and $^{13}$C NMR spectroscopy, elemental analysis, and single-crystal X-ray diffraction analysis.

#### 2.2.1. ESI-MS and NMR Spectroscopy

The identity of the obtained oxo-bridged W(V) dimers was verified by ESI-HRMS spectra, measured in the negative ion mode. Isotopic distribution patterns as well as *m/z* values were in good agreement with calculated data for the $[M-H^{+}]^{-}$ and $[M+HCOO^{-}]^{-}$ ions (Figures S3–S6). In contrast to the broad signals observed in the $^{1}$H NMR spectra of $Na_{2}[W_{2}O_{4}(EDTA)]$ at RT [18,25], rather resolved signals could be detected in the case of its neutral analogs (**2–4**), suggesting more rigid structures. The spectra revealed a double set of signals, as exemplified by complex **3** in Figure 3, which can be explained with the presence of two isomers in solution; integration suggests a ratio of 1:0.8. The spectral pattern does not change with increasing the temperature (Figure S7) or changing the NMR solvent (Figure S8). Indeed, the tertiary N atoms in complexes **2–4** become chiral and thus different stereoisomers (2 enantiomers and 2 diastereomers) could be formed. In the case of L2, two species (**2a** and **2b**), identified as $[W_{2}O_{4}(L2)]$ by ESI-HRMS and which display different $^{1}$H NMR spectra (Figure 4), could be isolated. Separation of **2a** and **2b** was possible due to slight differences in their solubilities. X-ray crystallographic analysis allowed to determine one of the species as the *cis* (a racemate of the *R,S/S,R* enantiomers) isomer (*vide infra*).

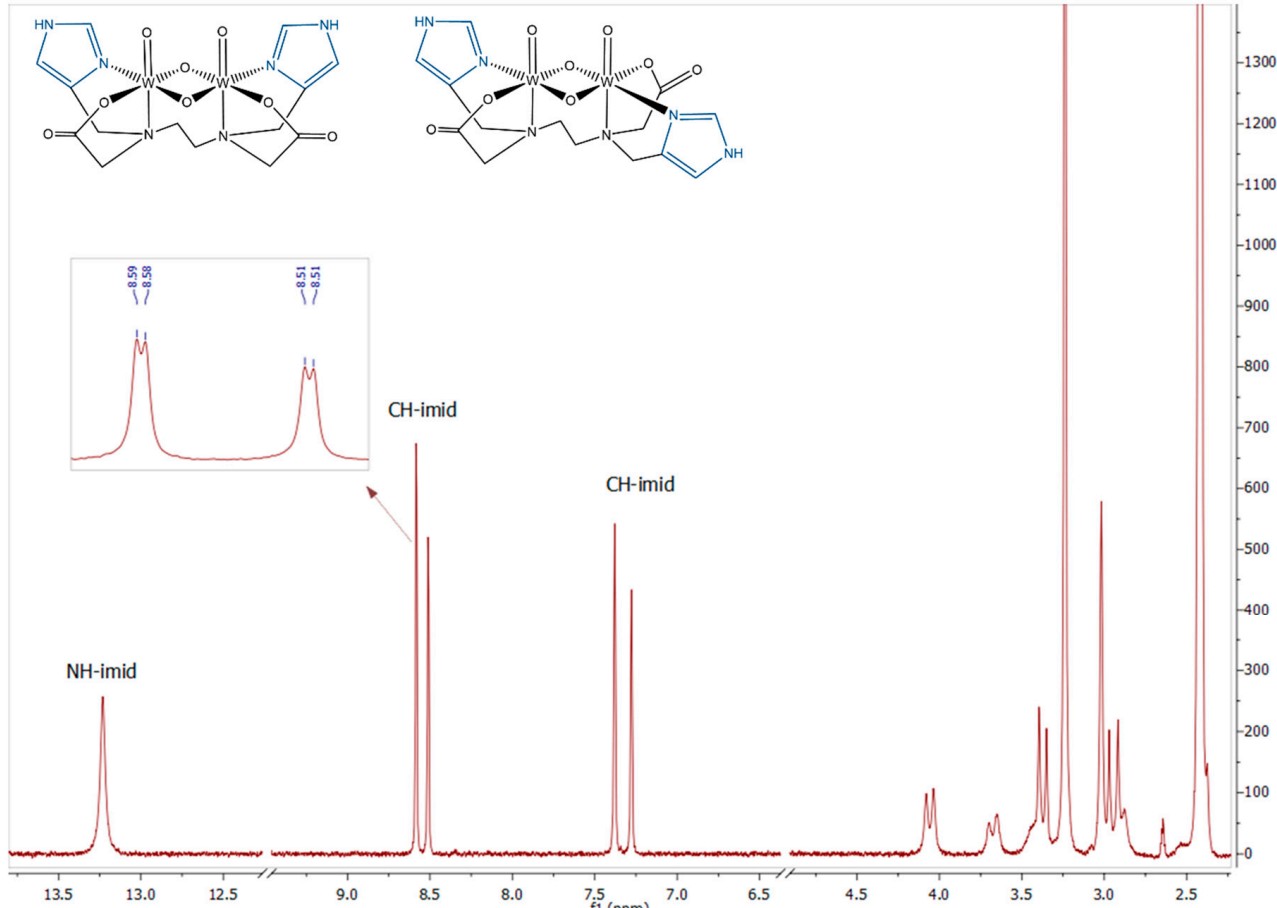

**Figure 3.** $^{1}$H NMR spectra of $[W_{2}O_{4}(L3)]$ (**3**) in DMSO-d$_{6}$ at RT; chemical formulae of two possible isomeric forms of the compounds are also shown. Peak assignment in the region below 4 ppm is difficult due to the several ethylene and methylene moieties presented in the molecule resulting in a complex pattern.

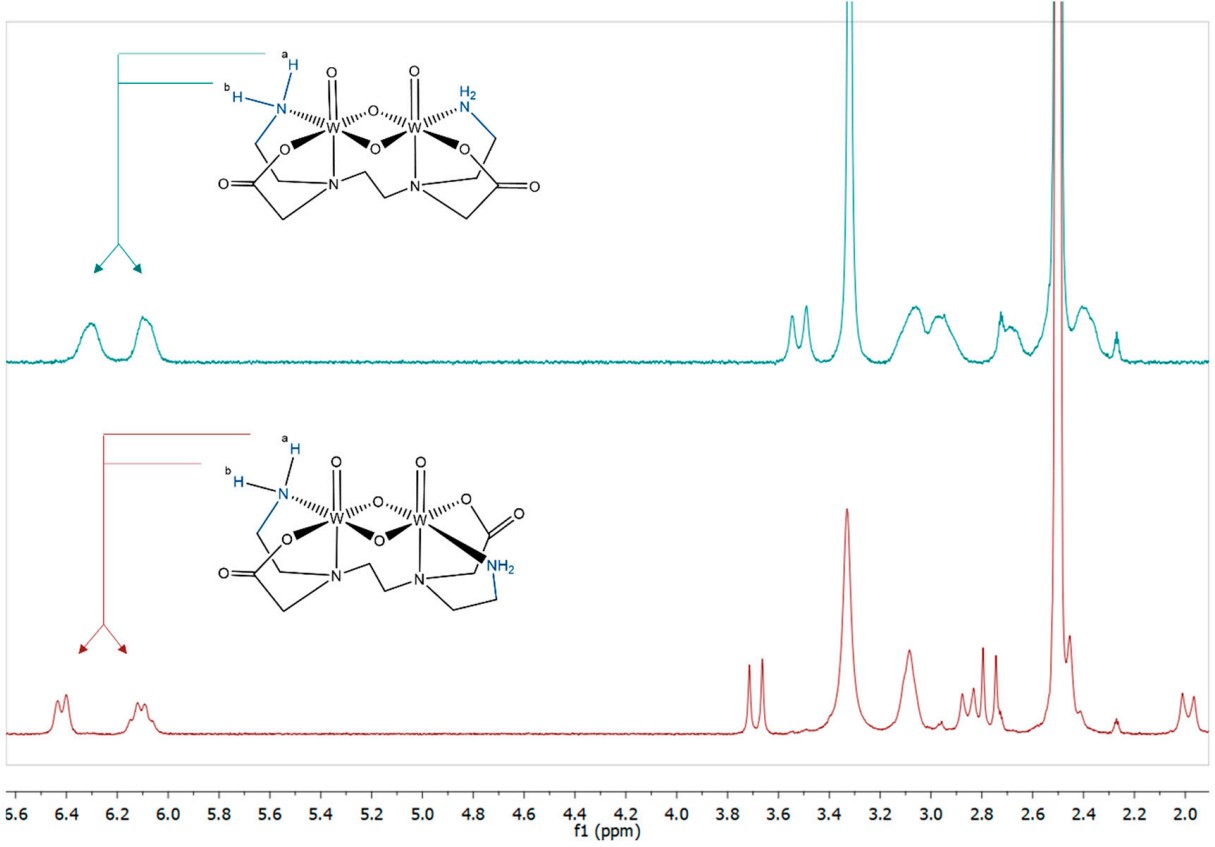

**Figure 4.** $^1$H NMR spectra of both isomers of [W$_2$O$_4$(L2)] (**2a** top and **2b** bottom) in DMSO-d$_6$ at RT; the structure of the *cis* isomer (**2a**) was confirmed by X-ray crystallographic analysis. The two NH$_2$ protons in both **2a** and **2b** are chemically non-equivalent, and thus give rise of two signals in the region of 6.0–6.6 ppm (see also Figure S9). The spectra in the region below 4 ppm display a complex pattern as a result of the several ethylene and methylene moieties present in the molecule.

### 2.2.2. Crystal Structures

The molecular structures of [W$_2$O$_4$(L2)] (**2a**), [W$_2$O$_4$(L3)] (**3**), [W$_2$O$_4$(L4)] (**4**), and [W$_2$O$_4$(His)$_2$] (**5**) were determined by single-crystal X-ray diffraction analysis. Suitable single crystals were obtained from aqueous or DMSO solutions of the compounds after several days of storage at 4 or 20 °C. Molecular views of **2a** and **3** are shown in Figure 5, and of complexes **4** and **5** in Figure 6. Important bond lengths and angles are listed in Tables 1 and 2. Crystal data, data collection parameters, and structure refinement details are given in the Supplementary Materials. Compounds **2a**, **3,** and **5** crystallized as hydrates (with 2 or 3 molecules of water) in the monoclinic space group P2$_1$/n (**2a**), orthorhombic space group Pna2$_1$ (**3**), and the trigonal space group P3$_2$21 (**5**), respectively. In the case of **3**, the complex molecules are arranged around tubes parallel to the a axis filled with water molecules. The latter form chains whereby each O atom is bound to the next of the chain via a hydrogen bond and, on the other hand, a hydrogen bond is built to an oxo atom of a complex (Figure S11, Table S6). Complex **4** crystallized as a DMSO solvate in the tetragonal space group I4$_1$ with asymmetric unit consisting of a half [W$_2$O$_4$(L4)] molecule and two DMSO solvent molecules (see Figure S12).

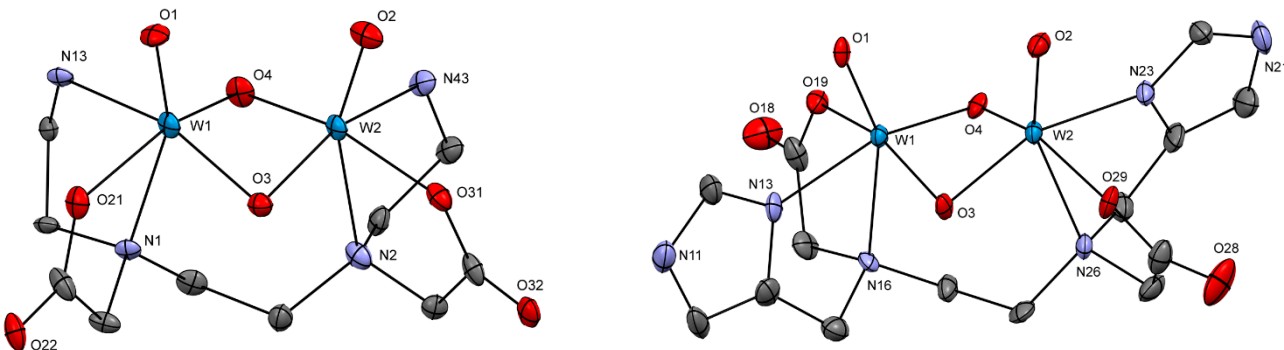

**Figure 5.** ORTEP view of complexes **2a** (**left**) and **3** (**right**) with atom labeling scheme. The thermal ellipsoids have been drawn at 50% probability level. Hydrogen atoms as well as solvent molecules are omitted for clarity.

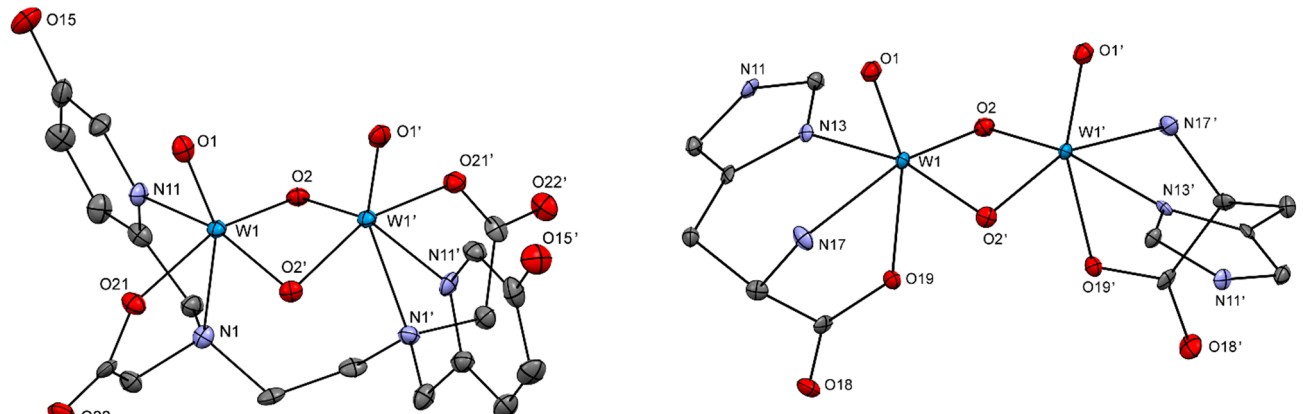

**Figure 6.** ORTEP view of complexes **4** (**left**) and **5** (**right**) with atom labeling scheme. Symmetry related atoms are labeled with X′ using the following symmetry transformations: -x, 1-y, z in **4,** and -x, y-x, 5/3-z in **5**. The thermal ellipsoids have been drawn at 50% probability level. Hydrogen atoms as well as solvent molecules are omitted for clarity.

**Table 1.** Selected bond lengths (Å) and angles (°) for complexes **2a**, **3**, **4**, and Ba[$W_2O_4$(EDTA)] [26].

| | **2a** | **3** | **4** | **Ba[$W_2O_4$(EDTA)]** |
|---|---|---|---|---|
| **W … W** | 2.5657(10) | 2.5555(5) | 2.5606(6) | 2.542(2)–2.557(3) |
| **W=O** | 1.709(8)–1.722(9) | 1.712(6)–1.735(6) | 1.716(8) | 1.696(46)–1.746(51) |
| **W-O$_\mu$** | 1.938(8)–1.960(9) | 1.932(5)–1.946(6) | 1.946(6)–1.960(6) | 1.883(30)–1.957(30) |
| **W-N (W-NC$_2$H$_4$N)** | 2.391(9)–2.418(11) | 2.389(7)–2.412(8) | 2.362(8) | 2.469(40)–2.480(28) |
| **W-O (W-OCO)** | 2.072(9)–2.119(8) | 2.081(6)–2.066(6) | 2.086(7) | 2.032(31)–2.107(29) |
| **W-N** | 2.185(6)–2.224(5) | 2.171(8)–2.156(8) | 2.212(8) | - |
| **O=W-N (W-NC$_2$H$_4$N)** | 156.5(4)–157.6(4) | 155.4(3)–157.7(3) | 154.6(3) | 151.8(3)–161.5(3) |
| **O$_\mu$-W-O$_\mu$** | 92.0(4)–92.3(4) | 91.8(2)–92.1(3) | 92.84 | 89.0(4)–93.4(8) |
| **W-O$_\mu$-W** | 81.8(3)–82.6(3) | 82.1(2)–82.6(2) | 81.9(2) | 81.4(2)–85.6(5) |
| **O$_\mu$-W-O$_\mu$-W** | 25.0(4) | 25.3(4) | 24.1(5) | 23.2(7)–25.0(7) |

In all complexes, the two W atoms are connected to each other via two bridging O atoms building a non-planar four-membered rings with O–W–O–W torsion angles, that are in good agreement with those reported for Ba[$W_2O_4$(EDTA)] [26] (Table 1). The measured short W-W distances (2.5555(5)–2.5657(10) Å) are similar to those observed for other

complexes featuring the $W_2O_4$ core and suggest a single metal-metal bond [17,26–28]. The distorted octahedral coordination spheres of the W atoms are completed by the hexadentate ligand L2-L4 (complexes **2a**, **3**, and **4**) or by two L-histidinates coordinated in a tridentate mode (complex **5**). The N atoms of the central ethylene diamine unit in complexes **2a**, **3**, **4** and Ba[$W_2O_4$(EDTA)] [26] are in *trans* positions to the W=O oxygen atoms. Consequently, the respective W-N bonds (2.362(8)-2.480(28) Å) are distinctively longer than the W-OCO (2.032(31)-2.119(8) Å) and W-N bonds (2.156(8)-2.224(5) Å) where the O and N atoms are *trans* to the μ-oxo atoms (Table 1). In the case of complex **5,** the O atoms of the COO groups are in *trans* position to the W=O units while the amino groups and the N atoms of the imidazole rings are *trans* to the bridging O atoms. The W-O and W-N bond lengths are comparable with those observed in the bis(μ-sulfido) analog [20] of **5** (Table 2).

**Table 2.** Selected bond lengths (Å) and angles (°) for complex **5** and its di-μ-sulfido analog [20].

| | [$W_2O_4$(His)$_2$] (5) | [$W_2O_2S_2$(His)$_2$] |
|---|---|---|
| **W=O** | 1.737(4) | 1.692(11)–1.722(10) |
| **W-O$_\mu$/W-S$_\mu$** | 1.946(4) | 2.313(4)–2.348(4) |
| **W-O (W-OCO)** | 2.139(4) | 2.137(10)–2.203(10) |
| **W-N (NH$_2$)** | 2.227(3) | 2.224(11)–2.239(12) |
| **W-N (imid)** | 2.184(5) | 2.218(11)–2.240(11) |
| **O=W-O** | 158.50(18) | 156.5(4)–161.9(5) |
| **O$_\mu$-W-O$_\mu$/S$_\mu$-W-S$_\mu$** | 95.40(17) | 102.15(14)–104.29(14) |
| **W-O$_\mu$-W/W-S$_\mu$-W** | 82.04(16) | 74.10(12)–75.27(13) |
| **O$_\mu$-W-O$_\mu$-W/S$_\mu$-W-S$_\mu$-W** | 17.14 | 12.08(16)–18.56(14) |

Complex **2a** crystallized in a centrosymmetric space group as a racemate of the *R,S* and *S,R* enantiomers with the two carboxylates, respectively two NH$_2$ groups, in *cis* positions to each other. In contrast, the COO groups and the N atoms from the imidazole (respectively pyridine) ring were in *trans* positions to each other in the crystal structures of complexes **3** and **4**. However, NMR spectra suggested that both diastereomers of **3** and **4** are present in the solution.

### 2.3. Solubility and Stability

Neutral oxo-bridged W(V) dimers exhibited only limited solubility in water (~1 mg/mL for **2b** and **3**, and <0.5 mg/mL for **2a**, **4** and **5**) and common organic solvents, except for complex **3**, which is well soluble in DMSO. All complexes except for **5** showed to be stable in DMSO and aqueous solutions, as evident by RP-HPLC and $^1$H NMR measurements over a period of a few days (Figure S14). Furthermore, the stability of complex **3**, at physiological pH was examined by means of UV-Vis spectroscopy kinetic measurements. No signs of decomposition could be observed over 20 h of incubation at 25 °C (see Figure S15). Nevertheless, the compound possesses insufficient solubility for further development as contrast agent candidate.

### 3. Materials and Methods

All reagents and solvents were obtained from commercial suppliers and used without further purification. High-purity water used in the synthetic procedures and HPLC experiments was obtained from a Milli-Q water purification system (Millipore GmbH, Vienna, Austria). Microwave-assisted experiments were carried out in a Biotage® Initiator + (Biotage AB, Uppsala, Sweden) single-mode microwave instrument producing controlled irradiation at 2.45 GHz in Pyrex microwave reaction vials. The vials were equipped with magnetic stirring bars, and magnetic stirring at a rate of 720 rpm was used throughout the experiments. Reaction times refer to hold times at the temperatures indicated, not to total irradiation times. The temperature was measured with an IR sensor on the outside

of the reaction vessel. HPLC analyses were performed on Agilent 1200 series, equipped with an Agilent Zorbax Eclipse XDB-C18 (4.6 × 150 mm, 5 μm) column at 25 °C, a flow rate of 1.0 mL min$^{-1}$, and UV-Vis detection. Mobile phase consisted of 0.1% Bu$_4$NCl in water/MeOH mixtures. High-resolution mass spectra (HRMS) were obtained on an Agilent Technologies 6230 TOF LC/MS instrument equipped with an ESI ion source (negative ionization mode); acetonitrile/water (40: 60) mixture with 0.1% 5 M HCOONH$_4$ was used as a solvent. NMR spectra were recorded on a Bruker Avance III 300 MHz spectrometer in DMSO-d$_6$ or D$_2$O at an ambient temperature; in certain cases, various temperature measurements were also performed. Chemical shifts δ were referenced with the residual solvent peaks of the respective NMR solvents used and are given in ppm. J values are given in Hz. The multiplicity of peaks is denoted as broad signal (bs), singlet (s), doublet (d), or multiplet (m). NMR and HRMS data were processed using MestreNova. Elemental analyses (C, H, N, S) were performed at the Department of Inorganic Chemistry at the University of Technology in Graz using a Heraeus Vario Elementar automatic analyzer, and are within ±0.4% of the calculated values, confirming ≥95% purity of the respective compounds. UV-Vis spectroscopy scanning kinetic measurements (for stability studies) were performed on a Varian Cary® 50 spectrophotometer equipped with VWR 1140S recirculating thermostat.

### 3.1. Synthesis

Synthesis and characterization of ligands L1-4 is described elsewhere [20]. L-histidine and DL-2,3-diaminopropionic acid were purchased from commercial vendors.

A general scheme for the synthesis of the new oxo-bridged W(V) dinuclear complexes is given in Scheme 1. The oxalato precursor (**P1**) was prepared according to literature [16,21] with minor revisions. The molecular formula of the compound (i.e., the amount of oxalate) was determined by elemental analyses for every individual batch in order to provide consistent stoichiometries for subsequent reactions.

General Procedure for the Synthesis of Neutral Complexes of the Type [W$_2$O$_4$(L)], L = L2–4, and [W$_2$O$_4$(His)$_2$]

An aqueous solution of the respective ligand (L2-4 or L-histidine, 1.5–5 equiv) was prepared in a 20 mL microwave reaction vial, and pH was adjusted to c.a. 5.5–6 by the addition of 1 M NaOH or 1 M HCl (in the case of L-histidine). Subsequently, the solution was purged with a flow of N$_2$ for 10 min, and the tungsten oxalato precursor (**P1**, 1 equiv) was added. The microwave vial was then tightly closed under a flow of N$_2$, placed in the microwave instrument, and heated at 120–140 °C for 1.5–5 min. After cooling to 45 °C, a warm solution of calcium acetate (3 equiv) was added, and the reaction mixture was stirred under a gentle flow of N$_2$ at 50 °C for 10 min. The calcium oxalate formed was removed by vacuum filtration via sintered glass filter (P5), equipped with a filter paper disk (MN GF-5). The obtained clear solution was concentrated (by rotary evaporation) and placed at 4 °C for 6–20 h (in the case of **5** under N$_2$ atmosphere) to yield the desired product. The latter was collected via filtration, washed with minimal amounts of cold water and MeOH, and dried in vacuo.

*cis*-[W$_2$O$_4$(L2)]·4H$_2$O (2a)

H$_2$L2·4HCl (469 mg, 1.149 mmol, 5 equiv) in water (12 mL), 1 M NaOH (4.2 mL), **P1** (200 mg, 0.230 mmol, 1 equiv), 140 °C for 5 min; calcium acetate (120 mg, 0.758 mmol in H$_2$O (2 mL). Yield: 40 mg (45%), orange powder. Crystals suitable for X-ray diffraction analysis were obtained from an aqueous solution of **2a** after several days of storage at RT. $^1$H NMR (DMSO-d$_6$, 300 MHz) δ 6.30 (bs, 2H, NH$_2$), 6.11 (bs, 2H, NH$_2$), 3.52 (br d, *J* = 16.6 Hz, 2H), 3.07 (bs, 4H), 2.96 (bs, 2H), 2.70 (bs, 2H), 2.39 (bs, 2H), 1.99 (m, 2H) ppm. ESI-HRMS(-) found (calculated): *m/z* [M-H$^+$]$^-$, 691.0191 (691.0227); [M+HCOO$^-$]$^-$, 737.0269 (737.0282). Anal. calcd. for C$_{10}$H$_{20}$N$_4$O$_8$W$_2$·3H$_2$O: C, 16.10; H, 3.51; N, 7.51. Found. C, 15.80; H, 3.15; N, 7.18.

*trans*-**[W$_2$O$_4$(L2)]·3H$_2$O (2b)** was isolated from the filtrate of **2a** after further reducing the volume and storage at 4°C for 1 week. The orange precipitate obtained was collected via filtration, washed with minimal amounts of cold water and MeOH, and dried in vacuo. Yield: 24 mg (27%), orange powder. $^1$H NMR (DMSO-d$_6$, 300 MHz) δ 6.42 (m, 2H, NH$_2$), 6.11 (m, 2H, NH$_2$), 3.69 (d, *J* = 15.3 Hz, 2H), 3.08 (bs, 5H), 2.85 (d, *J* = 13.2 Hz, 2H), 2.77 (d, *J* = 15.3 Hz, 2H), 2.45 (overlapped with the solvent peak), 1.99 (d, *J* = 13.6 Hz, 2H) ppm. $^{13}$C NMR (DMSO-d$_6$, 75 MHz): δ 174.1 (COO), 60.3, 60.2, 50.2, 38.7 (under the residual solvent peak) ppm. ESI-HR MS(-) found (calculated): *m/z* [M-H$^+$]$^-$, 691.0226 (691.0227). Anal. calcd. for C$_{10}$H$_{20}$N$_4$O$_8$W$_2$·4H$_2$O: C, 15.72; H, 3.69; N, 7.33. Found. C, 15.62; H, 3.32; N, 6.97.

**[W$_2$O$_4$(L3)]·2H$_2$O (3)**

H$_2$L3·4HCl (185 mg, 0.386 mmol, 1.5 equiv) in water (10 mL), 1 M NaOH (1.4 mL), **P1** (200 mg, 0.258 mmol, 1 equiv), 120 °C for 1.5 min; calcium acetate (108 mg, 0.683 mmol in H$_2$O (2 mL). Yield: 160 mg (78%), orange crystals (suitable for X-ray data collection). $^1$H NMR (DMSO-d$_6$, 300 MHz) δ 13.32 (bs, 2H, imid-NH), 8.67 and 8.59 (d + d, *J* = 1.1 Hz, 1H, imid-CH), 7.46 and 7.36 (s + s, 1H, imid-CH), 4.15 (m, 1H), 3.75 (m, 1H), 3.46 (m, 1H), 3.16 (s, 2H), 3.00 (m, 2H), 2.60 (m, 1H), 2.46 (overlapped with solvent residual signal) ppm. $^{13}$C NMR (DMSO-d$_6$, 75 MHz): δ 173.2 (COO), 138.9 and 138.5 (imid-CH), 135.8 and 135.4 (imid-CH), 115.1 (imid-CH), 63.1, 55.2, 52.4 ppm. ESI-HRMS(-) found (calculated): *m/z* [M-H$^+$]$^-$, 765.0141 (765.0132). Anal. calcd. for C$_{14}$H$_{18}$N$_6$O$_8$W$_2$·2H$_2$O: C, 20.97; H, 2.76; N, 10.48. Found. C, 21.03; H, 2.54; N, 10.62.

**[W$_2$O$_4$(L4)]·3H$_2$O (4)**

H$_2$L4·4HCl (172 mg, 0.320 mmol, 1.4 equiv) in water (16 mL), 1 M NaOH (1.3 mL), **P1** (200 mg, 0.230 mmol, 1 equiv), 140 °C for 3 min; calcium acetate (110 mg, 0.695 mmol) in H$_2$O (2 mL). Yield: 30 mg (15%), orange powder. Crystals suitable for X-ray diffraction analysis were obtained from a DMSO-d$_6$ solution of **4** after a few days of storage at RT. $^1$H NMR (DMSO-d$_6$, 300 MHz) δ 10.98 (bs, 2H, py-OH), 8.97 (bs, 2H, py), 7.68 and 7.60 (m, 4H, py), 4.50 (bs, 1H), 3.80 (m, 2H), 3.25 (s, 3H), 2.60 (overlapped with solvent residual signal) ppm. ESI-HRMS(-) found (calculated): *m/z* [M-H$^+$]$^-$, 819.0115 (819.0125). Anal. calcd. for C$_{18}$H$_{20}$N$_4$O$_{10}$W$_2$·3H$_2$O: C, 24.73; H, 3.00; N, 6.41. Found. C, 24.47; H, 2.79; N, 6.31.

**[W$_2$O$_4$(His)$_2$]·3.8H$_2$O (5)**

L-Histidine (160 mg, 1.031 mmol, 4 equiv) in water (10 mL), 1 M HCl (0.35 mL), **P1** (200 mg, 0.258 mmol, 1 equiv), 140 °C for 3 min; calcium acetate (109 mg, 0.688 mmol in H$_2$O (2 mL). Yield: 96 mg (46%), orange crystals (suitable for X-ray data collection). ESI-HRMS(-) found (calculated): *m/z* [M-H$^+$]$^-$, 738.9969 (738.9981). Anal. calcd. for C$_{12}$H$_{16}$N$_6$O$_8$W$_2$·3.8H$_2$O: C, 17.83; H, 2.94; N, 10.40. Found. C, 17.45; H, 2.49; N, 10.25.

*3.2. Crystallographic Structure Determination*

Single-crystal X-ray diffraction measurements were performed on a Bruker APEX-II CCD diffractometer using monochromatized Mo-Kα radiation at 100 K. Molecular structures were solved by direct methods (SHELXS-97) [29] and refined by full-matrix least-squares techniques against F2 (SHELXL-2014/6) [30]. Further details, including crystal data, data collection parameters, and structure refinement can be found in the Supplementary Materials. Additional information can be obtained from the Cambridge Crystallographic Data Center (CCDC 2239128-2239131).

**4. Conclusions**

A series of new neutral di-μ-oxo W(V) dinuclear complexes featuring L-histidine or disubstituted EDTA derivatives as ligands have been synthesized via microwave-assisted ligand exchange reaction at pH 5.5–6. The use of microwave heating allowed short reaction times and suppressed possible side product formation associated with the decomposition of the W(V) oxalato precursor. The obtained tungsten compounds were characterized in detail by ESI-HRMS, NMR spectroscopy, elemental analysis, and X-ray crystallography. To the best of our knowledge, these are the first examples of non-ionic complexes, that

contain the $W_2O_2(\mu\text{-}O)_2$ core and *N,O*-chelating ligands. Such compounds could be of high interest as candidates for X-ray imaging agents due to their high tungsten content (>44%) and non-ionic nature. Unfortunately, the herein investigated complexes exhibited poor aqueous solubility, and thus their potential utility as CT contrast media could not be further evaluated.

**Supplementary Materials:** The following supporting information can be downloaded at: https://www.mdpi.com/article/10.3390/inorganics11030114/s1, Figure S1: Main peaks observed in the ESI-HRMS spectra of the oxalatotungstate(V) complex **P1**; Figure S2: Stability of aqueous solutions of complex **P1** followed by UV-Vis spectroscopy; Figures S3–S6: ESI-HRMS spectra of complexes **3**, **4**, **2b** and **5**; Figure S7: Variable temperature $^1$H NMR spectra of complex **3** in DMSO-d$_6$; Figure S8: $^1$H NMR spectra of complex **3** in D$_2$O; Figure S9: COSY of **2b** in DMSO-d$_6$; Crystal structure determination; Figure S10: Stereoscopic ORTEP plot of the asymmetric unit of **2a** showing the atomic numbering scheme.; Figure S11: Stereoscopic ORTEP plot of **3** showing the atomic numbering scheme.; Figure S12: Stereoscopic ORTEP plot of complex **4** and the two DMSO molecules of the asymmetric unit showing the atomic numbering scheme.; Figure S13: Stereoscopic ORTEP plot of complex **5** showing the atomic numbering scheme.; Table S1: Crystallographic data and structure refinement for complexes **2a** and **3**; Table S2: Crystallographic data and structure refinement for complexes **4** and **5**; Table S3: Selected bond lengths [Å] and angles [°] for **2a**; Table S4: Hydrogen bonds for **2a** [Å, °]; Table S5: Selected bond lengths [Å] and angles [°] for **3**; Table S6: Hydrogen bonds for **3** [Å, °]; Table S7: Selected bond lengths [Å] and angles [°] for **4**; Table S8: Hydrogen bonds for **4** [Å, °]; Table S9: Selected bond lengths [Å] and angles [°] for **5**; Table S10: Hydrogen bonds for **5** [Å, °]; Figure S14: Stability of complex **3** in DMSO-d$_6$; Figure S15. Stability of complex **3** dissolved in phosphate buffer (pH 7.4) as followed by UV-Vis spectroscopy.

**Author Contributions:** Conceptualization, H.P.V., T.B., S.H. and N.C.M.-Z.; methodology, H.P.V.; investigation, H.P.V. and F.B.; resources, H.P.V., T.G., F.B., T.B., S.H. and N.C.M.-Z.; original draft preparation, H.P.V.; writing, review and editing, T.G., F.B., T.B., S.H. and N.C.M.-Z. All authors have read and agreed to the published version of the manuscript.

**Funding:** This research was supported financially by Bayer AG and the University of Graz.

**Data Availability Statement:** Not applicable.

**Acknowledgments:** We would like to thank Bernd Werner for NMR measurements and Philipp Marco Neu for ESI-HRMS measurements. We also thank Tanja Scheer and Philipp Wahlhuetter for their help in synthesizing some of the compounds.

**Conflicts of Interest:** The authors declare no conflict of interest.

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
