# Peer review of "Neutral W(V) Complexes Featuring the W2O2(µ-O)2 Core and Amino Acids or EDTA Derivatives as Ligands: Synthesis and Structural Characterization"

_inorganics, doi:10.3390/inorganics11030114_

Round 1
Reviewer 1 Report
This paper describes a series of neutral W(V) complexes featuring a W2O2(µ-O)2 core and amino acids or EDTA derivatives as ligands. The paper includes synthesis and single crystal structure determinations of four of the compounds.
The structures are somewhat routine showing a well-known structural motif and indeed predictable, but they have been well-refined. The cif files (and checkcif files) are satisfactory.
In the spacegroups described in the text, appropriate subscripts should be used. Also all dimensions the text should be given their standard deviations.
In Figure 4, the symmetry related atoms should not be labelled. This would make it clear that the molecules have imposed symmetry. The nature of the symmetry in both molecules should be included in the caption and mentioned in the text.
In Table 1 the W…W distances should be included. These distances are short and it should be discussed as to whether they should be considered as bonds with references to the literature.
The nature of the solvent molecules in the four structures should be mentioned in the text in more detail. They are detailed in the Supplementary in satisfactory fashion but a brief account should be included in the text for each structure.
In addition to the crystallography contains characterization by ESI-HRMS, 1H and 13C NMR spectroscopy, and elemental analysis which are of some interest.
While the structures will be of limited interest, the paper is well written and should be published after minor revision taking into account the above comments.
Author Response
Thank you very much for the valuable comments. A point-by-point response is given below:
In the spacegroups described in the text, appropriate subscripts should be used. Also all dimensions the text should be given their standard deviations.
Answer: Have been corrected accordingly.
In Figure 4, the symmetry related atoms should not be labelled. This would make it clear that the molecules have imposed symmetry. The nature of the symmetry in both molecules should be included in the caption and mentioned in the text.
Answer: There seems to be a mix-up: Figure 4 does not show molecules with symmetry-related atoms, but Figure 6 does. In Figure 6, the symmetry related atoms are now labeled with X’ and the used symmetry transformation is mentioned in the Figure caption.
Nevertheless, Figure 4 was modified: the non-equivalence of the two NH2 protons in both isomers is emphasized. Accordingly, an additional sentence was added to the Figure caption.
In Table 1 the W…W distances should be included. These distances are short and it should be discussed as to whether they should be considered as bonds with references to the literature.
Answer: The respective distances were included in Table 1 and an additional sentence has been added to the text.
The nature of the solvent molecules in the four structures should be mentioned in the text in more detail. They are detailed in the Supplementary in satisfactory fashion but a brief account should be included in the text for each structure.
Answer: The number of co-crystallized solvent molecules was given in the text; two additional sentences regarding the crystal structure of complex 3 were also added.
Reviewer 2 Report
The manuscript describes preparation of new binuclear complexes of W(VI) with rather "classical" organic ligands. This is very surprising since the compounds of this family are well known for many years so it coule be expected that those presented in this work are known as well - however, they aren't. Therefore, this is a nice addition to the old good coordination chemistry of tungsten which is also relevant to catalysis and some other application areas. Technically, the work is performed very well, so I cannot give any critical comments.
My only recommendation is related to references. There are some papers which, in my opinion, would improve the introduction. This is not obligatory so authors can feel free to add on not to add these corrections. Otherwise, the manuscript can be accepted as it is.
1. Catalytic activity of W(VI) compounds. There are some POM works which could be worth mentioning:
a) 10.1039/c2cc31692g
b) 10.1039/c4cc09271f
2. Binuclear complexes of W2O4 family:
a) 10.1021/ic00285a020
b) 10.1016/j.ica.2014.09.026
c) 10.1016/S0020-1693(00)92347-8
d) 10.1246/bcsj.68.456
Author Response
Thank you very much for the suggested references. We have included a few of them:
1) 10.1039/c2cc31692g was cited in the introduction
2) 10.1016/j.ica.2014.09.026 and 10.1246/bcsj.68.456 were cited in the crystal structure discussion.